# Fluoroquinolone Metalloantibiotics: A Promising Approach against Methicillin-Resistant *Staphylococcus aureus*

**DOI:** 10.3390/ijerph17093127

**Published:** 2020-04-30

**Authors:** Mariana Ferreira, Lucinda J. Bessa, Carla F. Sousa, Peter Eaton, Dafne Bongiorno, Stefania Stefani, Floriana Campanile, Paula Gameiro

**Affiliations:** 1REQUIMTE-LAQV (Rede de Química e Tecnologia – Laboratório Associado para a Química Verde), Departamento de Química e Bioquímica da Faculdade de Ciências da Universidade do Porto, Rua do Campo Alegre, s/n, 4169-007 Porto, Portugal; lucinda.bessa@fc.up.pt (L.J.B.); up200907323@fc.up.pt (C.F.S.); peter.eaton@fc.up.pt (P.E.); agsantos@fc.up.pt (P.G.); 2Department of Biomedical and Biotechnological Sciences, University of Catania, Via Santa Sofia 97, 95123 Catania, Italy; d.bongiorno@unict.it (D.B.); stefanis@unict.it (S.S.); f.campanile@unict.it (F.C.)

**Keywords:** fluoroquinolones, metalloantibiotics, antimicrobial resistance, methicillin-resistant *Staphylococcus aureus*, atomic force microscopy

## Abstract

Fluoroquinolones (FQs) are antibiotics commonly used in clinical practice, although nowadays they are becoming ineffective due to the emergence of several mechanisms of resistance in most bacteria. The complexation of FQs with divalent metal ions and phenanthroline (phen) is a possible approach to circumvent antimicrobial resistance, since it forms very stable complexes known as metalloantibiotics. This work is aimed at determining the antimicrobial activity of metalloantibiotics of Cu(II)FQphen against a panel of multidrug-resistant (MDR) clinical isolates and to clarify their mechanism of action. Minimum inhibitory concentrations (MICs) were determined against MDR isolates of *Escherichia coli, Pseudomonas aeruginosa* and methicillin-resistant *Staphylococcus aureus* (MRSA). Metalloantibiotics showed improved antimicrobial activity against several clinical isolates, especially MRSA. Synergistic activity was evaluated in combination with ciprofloxacin and ampicillin by the disk diffusion and checkerboard methods. Synergistic and additive effects were shown against MRSA isolates. The mechanism of action was studied though enzymatic assays and atomic force microscopy (AFM) experiments. The results indicate a similar mechanism of action for FQs and metalloantibiotics. In summary, metalloantibiotics seem to be an effective alternative to pure FQs against MRSA. The results obtained in this work open the way to the screening of metalloantibiotics against other Gram-positive bacteria.

## 1. Introduction

The cell envelope is the first natural bacterial barrier against antimicrobials, comprising two major components, the cell membrane and the peptidoglycan [1,2]. In general, bacterial membranes are rich in anionic and zwitterionic lipids, especially phosphatidylglycerol (PG), cardiolipin (CL) and phosphatidylethanolamine (PE) [3,4], while peptidoglycan harbors sugars and several amino acids [5]. However, this structure differs between Gram-negative and Gram-positive bacteria. Gram-positive bacteria are surrounded by a single membrane, the cytoplasmic membrane, enriched with anionic phospholipids and proteins and a thicker peptidoglycan sheet (around 30–100 nm thickness), external to the cytoplasmic membrane [4,6,7]. In turn, Gram-negative bacteria comprise an inner membrane (cytoplasmic membrane) and an outer membrane, with different composition, separated by a thinner peptidoglycan layer (a few nm in thickness) [4,7]. The inner membrane is a phospholipidic bilayer mostly composed of zwitterionic and anionic phospholipids [7,8,9], while the outer membrane is an asymmetric bilayer that differs in composition between the inner and outer leaflets. The inner leaflet of the outer membrane is very similar to the inner membrane [10], while the outer leaflet is enriched with glycolipids (mainly lipopolysaccharides) [2,8,10,11] and proteins, such as lipoproteins or integral membrane proteins, known as outer membrane proteins (OMPs) or porins [9,12]. These structural differences explain the general high resistance of Gram-negative bacteria to most antimicrobial agents, compared to Gram-positive bacteria [10].

Fluoroquinolones (FQs) are a family of antibiotics with a large spectrum of action, affecting Gram-negative and some Gram-positive bacteria, and being widely used in clinical practice [13]. These antibiotics act intracellularly, inhibiting the activity of DNA gyrase and topoisomerase IV [14,15], and their translocation in Gram-negative bacteria is usually dependent on porins [16]. Nevertheless, due to their misuse and overuse, bacteria have developed several resistance mechanisms to FQs. Alterations to the target molecules (through chromosomal mutations or plasmid-acquired resistance genes) and the reduction of the intracellular concentration of the drugs (by decreasing the influx or increasing the efflux) are the main bacterial resistance mechanisms against FQs that have been reported [17,18]. One of the reported strategies to bypass bacterial resistance to FQs is their complexation with transition metals, and so far it has been shown that only copper(II) complexes are truly stable [19,20,21,22]. Furthermore, the addition of 1,10-phenanthroline (phen) gave rise to ternary complexes (CuFQphen) with high stability under physiological conditions (of concentration, pH and temperature) and antibacterial activity (comparable or improved, with regard to free FQs) [19,23,24]. These cationic molecules, known as metalloantibiotics, seem to be a promising strategy against Gram-negative bacteria due to their alternative influx route apparently independent of porins [19,21,23,25,26]. Consequently, these compounds may overcome some bacterial resistance mechanisms developed to FQs, being potential alternatives to pure FQs. Nonetheless, to our knowledge, studies of the effect of these metalloantibiotics against Gram-positive bacteria still scarce.

This work aims to evaluate the antimicrobial activity of five metalloantibiotics and their respective free FQs against clinical isolates of Gram-negative and Gram-positive bacteria, through the assessment of minimum inhibitory concentrations (MICs). Three bacterial species of clinical importance [27] (*Escherichia coli*, *Pseudomonas aeruginosa* and *Staphylococcus aureus*) were chosen and reference strains and multidrug-resistant (MDR) clinical isolates, including methicillin-resistant *S. aureus* (MRSA), were used. The compounds that revealed improved MIC values in comparison to pure FQs were also tested in combination with some antibiotics, through the disk diffusion and the checkerboard methods, to assess possible synergistic effects. To determine if they show a similar mechanism of action as FQS, the metalloantibiotics capacity to inhibit DNA gyrase and topoisomerase IV was evaluated through enzymatic inhibitory assays. Additionally, Atomic Force Microscopy (AFM) was used to evaluate the effect of FQs and metalloantibiotics on bacterial membrane features (shape and size).

## 2. Materials and Methods

### 2.1. Fluoroquinolones and Metalloantibiotics Preparation

Ciprofloxacin (cpx), enrofloxacin (erx), levofloxacin (lvx) and sparfloxacin (spx) (all >98.0%), were purchased from Sigma-Aldrich. Moxifloxacin (mxfx) was a gift from Bayer. All other chemicals were from Merck. Stock solutions of phen, Cu(II)/phen (1:1), Cu(NO_3_)_2_.3H_2_O, FQs (cpx, erx, lvx, mxfx and spx) and respective metalloantibiotics were prepared in 10 mmol dm^−3^ HEPES buffer (0.1 mol dm^−3^ NaCl; pH 7.4, using double deionized water), with the exception of the copper solution, prepared only in double deionized water. The solutions were filter-sterilized and stored in small aliquots, protected from light, at −80 °C. The stock solutions were then thawed as needed and used in the same day. The metalloantibiotic solutions used were prepared by mixing the three components (FQ, Cu(II) and phen) in stoichiometric proportions (1:1:1), as previously reported for other FQs metalloantibiotics [19,20,21,28,29,30]. The Cu(NO_3_)_2_.3H_2_O solution used to prepare the metalloantibiotics solutions was previously titrated in alkaline medium with EDTA and using murexide as indicator.

### 2.2. Antimicrobial Activity Assays

#### 2.2.1. Bacterial Strains and Culture Conditions

Four bacterial reference strains from ATCC (LGC Standards) were used: two Gram-negatives, *E. coli* ATCC 25922 and *P. aeruginosa* ATCC 27853 and two Gram-positives, *S. aureus* ATCC 25923 and ATCC 29213. Overall, 30 MDR clinical isolates were also tested: four MDR *P. aeruginosa* (from Portugal), eight MDR *E. coli* (from Portugal) and 18 MRSA (14 from Italy and four from Portugal). All the strains and isolates were stored in glycerol at −80 °C and thawed at room temperature before being plated on non-selective agar. The antibiotic resistance profiles of the clinical isolates are compiled in Appendix A, available as Appendix A.

#### 2.2.2. Determination of Minimum Inhibitory Concentrations (MICs)

The MIC values were determined for all tested compounds by the broth microdilution method, according to the recommendations of the Clinical and Laboratory Standards Institute (CLSI) [31]. Briefly, stock solutions were dispensed into 96-well plates and diluted with Mueller-Hinton broth (MHB, Sigma-Aldrich), in the case of clinical isolates from Portugal (assays done in Porto, Portugal), or cation-adjusted Mueller-Hinton broth (CAMH broth, Sigma-Aldrich), in the case of clinical isolates from Italy (assays done in Catania, Italy), using a two-fold dilution series. The highest in-test concentration was 1024 μg/mL. The test wells were inoculated with a final bacterial concentration of ≈ 5 × 10^5^ colony-forming units (CFU) per mL [31]. Control wells were performed for broth, inoculum and for each compound solution. The inoculated 96-well plates were incubated at 37 °C for 16 to 20 h. Afterwards, the MIC, the lowest concentration of an antimicrobial agent that completely inhibits the bacterial growth, detected by the naked eye [32], was determined. Three independent experiments were performed.

#### 2.2.3. Combination Synergy Study

The compounds that revealed improved activity against some clinical isolates of MDR *E. coli* and MRSA were tested in combination with other antibiotics in two-drug combination studies. Selected compounds were combined with cpx (5 μg) or ampicillin—amp (10 μg) against *E. coli* and MRSA isolates. The combined effect of compounds with antibiotics was tested by the disk diffusion and the checkerboard methods.

##### Disk Diffusion Method

A suspension of each selected MDR isolate was prepared in MHB by picking overnight colonies, and its turbidity was adjusted to OD_600_ = 0.1 (≈10^8^ CFU/mL). The suspension was then inoculated on the surface of a MH agar plate (150 mm diameter). Each plate housed up to three paper disks. Control disks (blank disks—6 mm diameter, Liofilchem) and commercial antibiotic-impregnated disks (cpx 5 μg and amp 10 μg, Oxoid) were loaded with the tested compound solution (15 µL of a 2048 μg/mL stock solution to reach 30 μg of each compound/per disk). The plates were then incubated at 37 °C, for 16 to 20 h. The diameter of the zones of growth inhibition observed around each disk was measured (in mm) [32]. Each compound was tested in duplicate and two independent experiments were performed. Potential synergism was considered when the halos of antibiotic disks impregnated with the tested compounds where greater compared to the halos of antibiotic disks or compound-impregnated disks alone.

##### Checkerboard Method

The compounds that exhibited potential synergism in the previous assay were also tested in two-drug combinations through the broth microdilution checkerboard method. MIC values were determined for each tested compound alone and in combination. The method was performed using a two-fold dilution series in 96-well plates and final bacterial concentration of ≈ 5 × 10^5^ CFU/mL [8], as previously described. The tested compound was diluted along the ordinate and the antibiotic along the abscissa. MIC values were determined after 16 to 20 h of incubation, at 37 °C [7,8]. Two independent experiments were performed. MIC values determined were used to calculate the fractional inhibitory concentration (FIC) of each compound/drug and the FIC index (ΣFIC), as follows: FIC of drug A (FIC A) = MIC of drug A in combination/MIC of drug A alone; FIC of drug B (FIC B) = MIC of drug B in combination/MIC of drug B alone; ΣFIC = FIC A + FIC B. The combinatorial effect was interpreted according to the following criteria: synergy, ΣFIC ≤ 0.5; additivity, 0.5 < ΣFIC ≤ 1; no interaction (indifference), 1 < ΣFIC ≤ 4; antagonism, ΣFIC > 4 [33].

### 2.3. Enzymatic Inhibitory Activity Assays

The study was carried out with two metalloantibiotics, Cucpxphen and Cuspxphen, and six compound concentrations were evaluated: 0.5; 1.0; 5.0; 10.0, 50.0 and 100.0 µmol dm^−3^. Gyrase supercoiling inhibition assays were performed by incubating relaxed pBR322 plasmid (0.5 μL per assay) with 1 unit of gyrase (prepared in the dilution buffer supplied) and the assay buffer supplied, in the absence and presence of a concentration range of the solutions being tested. Topoisomerase IV relaxation inhibition assays were performed by incubation of supercoiled pBR322 plasmid (0.5 μL per assay) with 1.5 (*E. coli*) or 2 (*S. aureus*) units of topoisomerase IV (prepared in the dilution buffer supplied) and the assay buffer supplied, in the absence and presence of a concentration range of the compounds tested. All the reactions were carried out according to the manufacturer’s instructions (Inspiralis, Norwich, UK). Three controls were used in the study: a negative control, consisting of a mixture of the plasmid, water, assay buffer and dilution buffer (in the absence of the enzyme), and two positive controls, comprising a mixture of the plasmid, water, assay buffer and water or HEPES buffer (in the presence of the enzyme). Cpx was used as a drug control in all experiments, due to its known enzymatic inhibitory activity. After each reaction, the plasmid (relaxed or supercoiled) was separated by agarose gel electrophoresis. All gels were stained with GreenSafe Premium (NZYTech), visualized and photographed under UV light. All determinations were performed in at least three independent experiments.

### 2.4. Evaluation of the Effect of Compounds on Bacterial Membranes by Atomic Force Microscopy

Bacterial cultures were grown in Luria broth (LB) until an optical density of about 0.1, corresponding to approximately 10^8^ CFU mL^−1^, was reached. Control samples (without antibiotic treatment) and samples containing FQ or metalloantibiotic in a concentration below the MIC (0.002–0.008 µg mL^−1^) were prepared. The study was carried out with two FQs (cpx and spx) and their metalloantibiotics, Cucpxphen and Cuspxphen. *E. coli* ATCC 25922 and *S. aureus* ATCC 25923 strains were used. Drops of 20 µL of each bacterial suspension (control and treated) were loaded onto a glass plate, followed by drying for 30 min at 37 °C, then gently washed with isotonic solution, subsequentially dried for 30 min at 37 °C and washed again with ultra-pure water and dried for 1 h at 37 °C. Samples were observed under a TT-AFM instrument from AFM Workshop with a 50 µm scanner in vibrating mode. Probes were purchased from Applied Nanostructures, Inc. (Mountain View, CA, USA), and had approximately 300 kHz resonant frequency, probe diameter <10 nm (AppNano ACT, Applied Nanostructures, Inc, Mountain View, CA, USA). Multiple areas per sample were imaged and representative images were selected. For *E. coli* experiments, three independent experiments, each composed by three independent replicates for each control and treated samples, were performed. For *S. aureus* experiments, however, due to time constraints, only one experiment composed by three independent replicates was performed. Nevertheless, at least 10 individual cells were analyzed in terms of height and size (length and diameter) in both experiments. Gwyddion software was used to process the images [34]. Statistical analysis was performed using GraphPad Prism^®^ software (GraphPad Software, San Diego, CA, USA).

## 3. Results and Discussion

### 3.1. Antimicrobial Activity against Multidrug-Resistant Clinical Isolates

The antibacterial activity of five free FQs (cpx, erx, lvx, mxfx and spx) and the respective metalloantibiotics of CuFQphen, phen, Cu(II)/phen (1:1) and Cu(NO_3_)_2_.3H_2_O salt was tested against bacterial reference strains and MDR clinical isolates.

There is a general practice to report MIC values in terms of ɣ (μg mL^−1^) [32,35]. Nevertheless, when comparing compounds with marked differences among their molecular weight (MW) values, the interpretation of the MIC results may be misleading when using ɣ. Therefore, as MW _Metalloantibiotics_ ≈ 2 × MW _FQs_ (Appendix A), the MIC results are presented in μmol dm^−3^, with all data available both in μg mL^−1^ and μmol dm^−3^ in the Appendix A.

The MIC values determined for all compounds against reference strains are presented in Table 1 and Appendix A. The MICs obtained for the free FQs meet the values recommended by the CLSI guidelines [35], with the exception of cpx against *P. aeruginosa* ATCC 27853, whose value (0.18 μmol dm^−3^, corresponding to 0.06 µg mL^−1^) was lower compared to the range recommended in the guidelines (0.25–1 µg mL^−1^). This result may arise from the absence of cation content in the MHB medium used. The CLSI guidelines refer that the cation content of the broth used in the antimicrobial susceptibility testing may affect the MIC results. Moreover, the supplementation of the broth with cations, in cases where lower MIC values were observed in *P. aeruginosa* ATCC 27853, are suggested in the guidelines [31]. Besides that, Chalkley and Koornhof have also published a value of 0.016 µg mL^−1^ for cpx against this control strain [36], which is even lower than the one obtained in this work.

Regarding metalloantibiotics, the MIC values were similar between each metalloantibiotic and the respective free FQ, which is in agreement with the data previously reported for *E. coli* ATCC 25922 by Feio et al. [19]. In turn, the MIC values obtained for phen, Cu(II)/phen (1:1) and copper solutions were considerably higher (more than 10^3^-fold higher) than those of FQs and metalloantibiotics. These outcomes were also previously stated by Feio et al. [19] and may help to confirm that metalloantibiotic dissociation does not happen, since the substituents by themselves did not show antimicrobial activity. The high MIC values of copper solutions were also previously reported by other authors [19,37,38].

Concerning MDR isolates, the MIC values obtained against Gram-negative bacteria were visibly different from those obtained against Gram-positive bacteria (Table 2 and Table 3 and Appendix A). The MICs obtained against all isolates tested are listed in full in the Appendix A. Some relevant excerpts are presented in Table 2 and Table 3.

MIC values could be obtained for all compounds, within the range of concentrations tested, against Gram-negative MDR isolates (Table 2, Appendix A), with the exception of the copper solution (MIC ≥ 4238.2 μmol dm^−3^). According to EUCAST guidelines, all the twelve MDR *E. coli* isolates tested were shown to be resistant to the tested FQs [39]. The MIC values of metalloantibiotics against MDR *E. coli* isolates (Table 2 and Appendix A) were comparable to those of the respective pure FQ. These results show that metalloantibiotics are not able to circumvent the bacterial resistance mechanisms to FQs of these clinical isolates. The MIC values of phen, Cu(II)/phen (1:1) and copper solutions were mostly similar to the ones previously obtained against the reference strains. Nevertheless, Cucpxphen revealed improved antibacterial activity compared to cpx against two clinical isolates, HSJ Ec002 and HSJ Ec003 (Table 2 and Appendix A). These results emphasize the importance of carefully choosing the units used to compare the antimicrobial activity of compounds with different molecular weights. In this particular case, the analysis of the results using conventional units (μg mL^−1^) suggests a difference of 2-fold (128 μg mL^−1^ for cpx and 64 μg mL^−1^ for Cucpxphen), while the comparison in μmol dm^−3^ discloses a difference of 4-fold (386.3 μmol dm^−3^ for cpx and 93.0 μmol dm^−3^ for Cucpxphen). Besides the improved antibacterial activity of Cucpxphen compared to free cpx, the MIC value is still quite high.

The antimicrobial activity of both metalloantibiotics and respective free FQs against MDR *P. aeruginosa* isolates was comparable (Appendix A), as evidenced by similar MIC values. The exceptions were observed for Cuerxphen, whose MIC values were higher compared to erx against two out of the four tested isolates. Together, the results obtained from *E. coli* and *P. aeruginosa* isolates suggest that metalloantibiotics may not bring any advantages over pure FQs against Gram-negative bacteria.

Interestingly, the results were more promising against *S. aureus* strains. The MIC values assessed against 18 MRSA clinical isolates are presented in Table 3 and Appendix A. Metalloantibiotics showed improved antibacterial activity in comparison to the respective pure FQs in 15 out of the 18 MRSA tested (Table 3 and Appendix A). MIC values of metalloantibiotics were 4 to 28-fold lower than those of FQs. Among the five metalloantibiotics, Cucpxphen and Cuspxphen exhibited effectiveness against a greater number of isolates. Only in three isolates (Sa4-SA3, 38/13 bis and 16/01) did pure FQs and the respective metalloantibiotics exhibited similar MIC values. Within the 18 MDR isolates tested, all strains were shown to be resistant to the tested FQs, according to EUCAST guidelines [39]. MIC values of phen and Cu(II)/phen (1:1) were comparable or greater than those of FQs and metalloantibiotics and no antimicrobial activity could be ascribed to the copper solution (MIC ≥3642.2 μmol dm^−3^). Despite the lower MIC values of metalloantibiotics compared to pure FQs, the absolute value of the MICs can still be considered rather high. Therefore, some synergistic assays were also performed to evaluate the effect of the combination of these metalloantibiotics with other antibiotics.

### 3.2. Combinatorial Effect of Metalloantibiotics with Antibiotics

According to the results obtained from MIC experiments, the metalloantibiotics that revealed improved activity against most clinical isolates of MDR *E. coli* and MRSA were tested in combination with some antibiotics (cpx and amp), in two-drug combinations. Cucpxphen and Cuspxphen were combined with cpx (5 μg) or amp (10 μg) against one *E. coli* isolate (HSJ Ec002) and two MRSA isolates (Sa1-SA3 and Sa3-SA3). The combination of phen, Cu(II)/phen (1:1) and Cu(NO_3_)_2_.3H_2_O salt with antibiotics was also evaluated.

Regarding *E. coli* isolate HSJ Ec002, a slight zone of inhibition (8 mm) for Cuspxphen in combination with cpx was observed, while each one alone caused no zone of inhibition (0 mm) (Appendix A). For all the other tested compounds, except phen, no growth inhibition zones were observed. The considerable inhibition halo obtained for phen against the *E. coli* isolate corroborates its lower MIC value previously obtained.

Concerning Gram-positive isolates, the diameter of zones of inhibition obtained are shown in Table 4. Cucpxphen, Cuspxphen and Cu(II)/phen (1:1) displayed defined zones of growth inhibition against MRSA isolate Sa1-SA3. The combination of Cucpxphen with amp revealed larger zones of inhibition. Furthermore, the combination of phen, Cu(II)/phen (1:1) or copper solutions with amp also induced considerable growth inhibition zones.

Regarding the MRSA isolate Sa3-SA3 (Table 4), a growth inhibition zone was caused by Cucpxphen, Cuspxphen, copper solution and amp. Cucpxphen in combination with the two tested commercial drugs (cpx and amp) led to a bigger zone of inhibition. The solutions of phen and Cu(II)/phen (1:1) caused no inhibition zones when alone or in combination. The inhibition zone of copper solution was not expected based on the high MIC value previously determined. However, its combination with amp increased the zone of bacterial inhibition. Based on the results achieved by the disk diffusion method, the work proceeded for the checkerboard method to evaluate the synergistic effect of Cuspxphen in combination with cpx against *E. coli* isolate HSJ Ec002 and of Cucpxphen in combination with amp and cpx against *S. aureus* isolate Sa1-SA3.

The checkerboard method allowed the calculation of the FIC index to further characterize the interactions among the different combinations (Table 5). The combination of Cuspxphen and cpx against the Gram-negative isolate revealed no change in the MIC value of the compounds when they were alone or in combination, indicating an indifferent interaction. However, the results were more encouraging against the two MRSA isolates, evidenced by the decrease in the MIC values of both compounds when in combination, disclosing a synergistic or additive interaction, depending on the extent of the effect, as revealed by the standard interpretation of FIC index values (Table 5).

Therefore, the strategy based on the combination of these metalloantibiotics with different antibiotics should be further explored. Nevertheless, when combining different compounds, it is important to understand their mechanism of action. Thus, this study finishes with enzymatic inhibitory assays and Atomic Force Microscopy studies in order to assess the mechanism of action of the metalloantibiotics.

### 3.3. Studies of the Mechanism of Action of the Metalloantibiotics

Although the mechanism of action of the metalloantibiotics is still unknown, their nuclease activity and the high affinity to intercalate into the DNA point to a mechanism similar to that of the free FQs [19,40]. For this reason, the enzymatic inhibitory activity of the metalloantibiotics was assessed against topoisomerases II (DNA gyrase) and IV of both *E. coli* and *S. aureus*. Two metalloantibiotics, Cucpxphen and Cuspxphen, and cpx (as a drug control due to its known inhibitory enzymatic activity) were tested. The enzymatic activity of cpx was always evaluated using two different concentrations (one able and one unable to inhibit the enzyme). The precise concentration (units—U) of enzyme required for each enzymatic inhibitory activity assay was previously determined (Appendix A and Appendix A). The activity data revealed that DNA gyrases are more active than topoisomerases IV (for the same amount of plasmid, 1 U of DNA gyrase is able to completely supercoil it, while 1.5 or 2 U of topoisomerase IV are necessary to totally relax it).

The results of the inhibitory enzymatic activity experiments, performed with cpx and the two metalloantibiotics studied, are presented in Table 6 and Figure 1 and Figure 2 (and Appendix A). Both metalloantibiotics showed inhibitory enzymatic activities similar to FQs, suggesting the same mechanism of action as FQs.

Comparing the results obtained from DNA gyrase supercoiling inhibition assays of *E. coli* and *S. aureus* (Table 6 and Figure 1 and Appendix A), the inhibitory activity of the three tested compounds was comparable within each assay. However, a higher concentration of compound was required to inhibit the same amount of enzyme in the case of the Gram-positive bacteria. This suggests that cpx and metalloantibiotics are more active against the DNA gyrase of the Gram-negative bacteria.

Regarding topoisomerase IV relaxation inhibition assays, differences between cpx and the metalloantibiotics were observed (Table 6 and Figure 2 and Appendix A). The concentration of metalloantibiotics required to inhibit topoisomerase IV was lower in comparison to cpx. This difference was more evident in the case of topoisomerase IV of *S. aureus*, showing that metalloantibiotics may be more effective against topoisomerase IV of Gram-positive bacteria.

Taking all results together, FQs and metalloantibiotics may easily target the DNA gyrase of *E. coli* and topoisomerase IV of *E. coli* and *S. aureus*. Furthermore, the DNA gyrase of *S. aureus* was shown to be less susceptible to the action of both compounds. Thus, it is expected that metalloantibiotics easily act on both topoisomerases in Gram-negative bacteria but have a preferential target, the topoisomerase IV, in the case of Gram-positive bacteria. Such outcomes are in accordance with several authors that refer to DNA gyrase as the major target of FQs in Gram-negative bacteria and topoisomerase IV as the preferential target in Gram-positive bacteria [17,18,41]. However, exceptions may occur in some bacterial species, depending on the FQ used. The higher activity of metalloantibiotics against topoisomerase IV of Gram-positive bacteria (compared to cpx) corroborates the improved antimicrobial activity herein observed against several MRSA isolates.

Taking into account the high affinity of metalloantibiotics to the lipid component of the membrane (high partition constants [19,28,42]), one could suppose that these compounds may affect or damage the membrane. To get evidence on this supposition, the interaction of FQs and metalloantibiotics with the bacterial membranes of *E. coli* and *S. aureus* was studied by AFM, upon bacterial cell treatment with sub-MIC concentrations of each compound.

AFM is a technique that uses the physical interaction of a sharp probe with the surface of the sample to build a map of the height of the sample’s surface, and is widely used to study micro-organisms, especially bacterial cells. Despite being slower and less suitable for cell-counting studies when compared to common optical microscopy methods, AFM has the great advantage of allowing high resolution three-dimensional images of the bacterial cell with clearer details of cell morphology and sub-cellular features under ambient conditions [43]. This technique allows the visualization of morphologic and mechanical changes in the bacteria upon treatment with antibacterial agents. Very small features in the bacterial membrane, such as holes, shrinking and cell shape changes can be revealed by AFM high resolution images. In this context, AFM was used to compare the effect of FQs or metalloantibiotics on bacterial membrane features, bacterial shape and size.

As membranes of Gram-negative and Gram-positive bacteria are largely distinct, we tested one model for each bacterial type: *E. coli* and *S. aureus*, respectively. Two FQs (cpx and spx) were chosen due to their different mechanism of permeation across the bacterial membrane: cpx translocation is strongly dependent on porins, while spx is expected to cross through the lipid bilayer [42]. Therefore, these two FQs and their respective metalloantibiotics were chosen to these experiments.

Appendix A show representative images of control samples of *E. coli* and *S. aureus* cells, respectively, and also depict an example of the plot of height and size measures analyzed in this study. For *S. aureus,* profile measurements show control cells with a height of about 500–700 nm and a diameter of about 1.0–1.5 µm (Appendix A). For *E. coli,* AFM images of control cells show a height ranging from 100–300 nm and a length of about 3–5 µm (Appendix A).

Analyzing the AFM images of treated bacterial cells (Figure 3 and Appendix A) it is clear that, apart from totally disrupted cells resulted from cell killing by the antibiotic action, it was not possible to see any special feature on the membrane caused by any of the compounds tested; no consistent holes or other features indicative of antibiotic action on the membrane were found. However, *E. coli* cells showed a clear filamentation after treatment with all the compounds (Figure 3). This pattern is confirmed by the plot depicted in Appendix A, where it is shown that the average length of the treated cells was about three times higher than the length of the untreated *E. coli* cells. This result is consistent with other published studies on *E. coli* morphology after treatment with cpx at a sub-MIC concentration [44,45]. Filamentation is a common response of some bacterial cells upon treatment with antibiotics that have a mechanism of action on the DNA or the DNA synthesis system, which is the case of FQs that inhibit DNA gyrase and topoisomerase. The filamentation is induced by a process known as the SOS response of the cell that inhibits cell division until the DNA can be repaired preventing the transmission of the damaged DNA [46]. Filamentation was observed in treated *E. coli* cells independently of the compound used in the treatment. No statistically significant differences in cell length were found upon treatment with the free FQ or the metalloantibiotics (Appendix A). Therefore, these results indicate that the mechanism of action of the metalloantibiotics in Gram-negative bacteria is similar to the one of the free FQs. It was also found that treatment with free FQ and CuFQphen did not change the height of the *E. coli* cells significantly (Appendix A). In this case, however, it is also important to mention that the correct determination of the cell height of the treated cells is hampered by the less clean background due to the presence of some artefacts like the one displayed in Figure 3B. This is caused by the presence of cell waste resulting from the leakage of cell contents, previously observed upon treatment with FQs [46]. Furthermore, bacteria with a collapsed appearance and “ghost cells” were found after treatment with FQ and metalloantibiotics, as are displayed in the treated cell images (Figure 3B–E). This effect of FQ treatment on bacterial cells was also reported in previous studies [46].

The antibiotics’ action on *S. aureus* cells produced slightly different results (Appendix A). The size of the cells was also increased after treatment with the antibiotics. However, the increment in cell size is less pronounced, when compared to the effect in *E. coli* (Appendix A). In fact, very few studies have reported alterations on *S. aureus* size upon treatment with FQs or other DNA targeting drugs. It has only been shown that gatifloxacin caused morphologic alterations on MDR *S. aureus* cells [47] and novobiocin causes variable modifications on *S. aureus* cell size [48]. Overall, we attribute the observed increase in size to abnormal cell division effects caused by the treatment with these compounds. Once again, no differences on cell size upon treatment with FQs and metalloantibiotics were found, pointing to a similar mechanism of action also on Gram-positive bacteria. Concerning cell height, very slight differences between treated and non-treated samples were found (Appendix A). In this case, only cells treated with Cucpxphen had a statistically different height than the control ones. It is important to report, however, that the analysis of the cell height of antibiotic-treated *S. aureus* samples was hampered by the presence of artefacts and an unclean background, while it was not the case for the control samples (representative images Appendix A). Nevertheless, it is possible to conclude that, once again, the effect of each FQ and its respective CuFQphen complex was similar, reinforcing an analogous mechanism of action.

## 4. Conclusions

MIC values revealed improved antimicrobial activity of metalloantibiotics in comparison to pure FQs, especially against MRSA isolates. Additionally, the combination of these complexes with cpx and amp exhibited synergistic or additive effects against MRSA isolates. Metalloantibiotics showed a similar mechanism of action to FQs, acting on topoisomerase IV and DNA gyrase of bacteria. Moreover, in the case of Gram-positive bacteria these complexes showed preference to act on topoisomerase IV. AFM analyses showed that metalloantibiotics cause no damage on bacterial membranes, allowing us to infer a probable intracellular mechanism of action.

As FQs are no longer used to fight infections caused by MRSA due to the bacterial resistance mechanisms developed against these drugs, the increased antimicrobial activity of metalloantibiotics against several MRSA clinical isolates may anticipate their potential use as alternatives to fight MDR Gram-positive bacteria.

## Figures and Tables

**Figure 1 ijerph-17-03127-f001:**
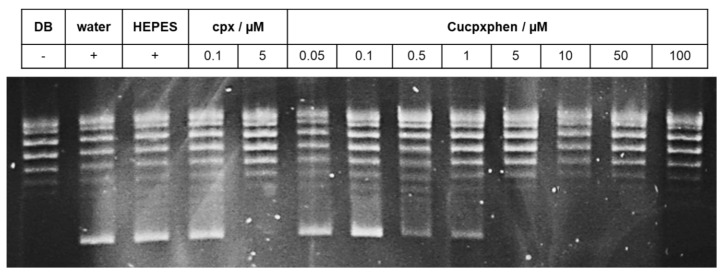
DNA gyrase supercoiling inhibition assay obtained for Cucpxphen as enzymatic inhibitor of the *E. coli* DNA gyrase, performed with 0.5 μL of relaxed pBR322 plasmid, determined in a 1% (w/v) agarose gel in TAE buffer. DB is dilution buffer and the respective band is the negative control, containing the relaxed plasmid in the absence of the enzyme. Water and HEPES are the positive control bands containing the enzyme and the plasmid. The cpx bands represent the drug control. µM means µmol dm^−3^ and refers to the concentration of the compound. The experiment was also performed with the *S. aureus* DNA gyrase supercoiling assay kit (Appendix A). The enzymatic inhibitory activity of Cucpxphen and Cuspxphen was evaluated for both bacterial species.

**Figure 2 ijerph-17-03127-f002:**
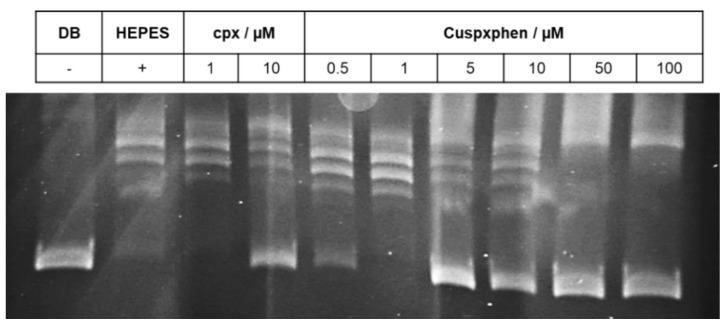
Topoisomerase IV relaxation inhibition assay obtained for Cuspxphen as enzymatic inhibitor of the *S. aureus* topoisomerase IV, performed with 0.5 μL of supercoiled pBR322 plasmid, determined in a 1% (*w*/*v*) agarose gel in TAE buffer. DB is dilution buffer and the respective band is the negative control, containing the relaxed plasmid in the absence of the enzyme. HEPES is the positive control containing the enzyme and the plasmid. The cpx bands represent the drug control. µM means µmol dm^−3^ and refers to the concentration of the compound. The experiment was also performed with the *E. coli* topoisomerase IV relaxation assay kit (Appendix A). The enzymatic inhibitory activity of Cucpxphen and Cuspxphen was evaluated for both bacterial species.

**Figure 3 ijerph-17-03127-f003:**
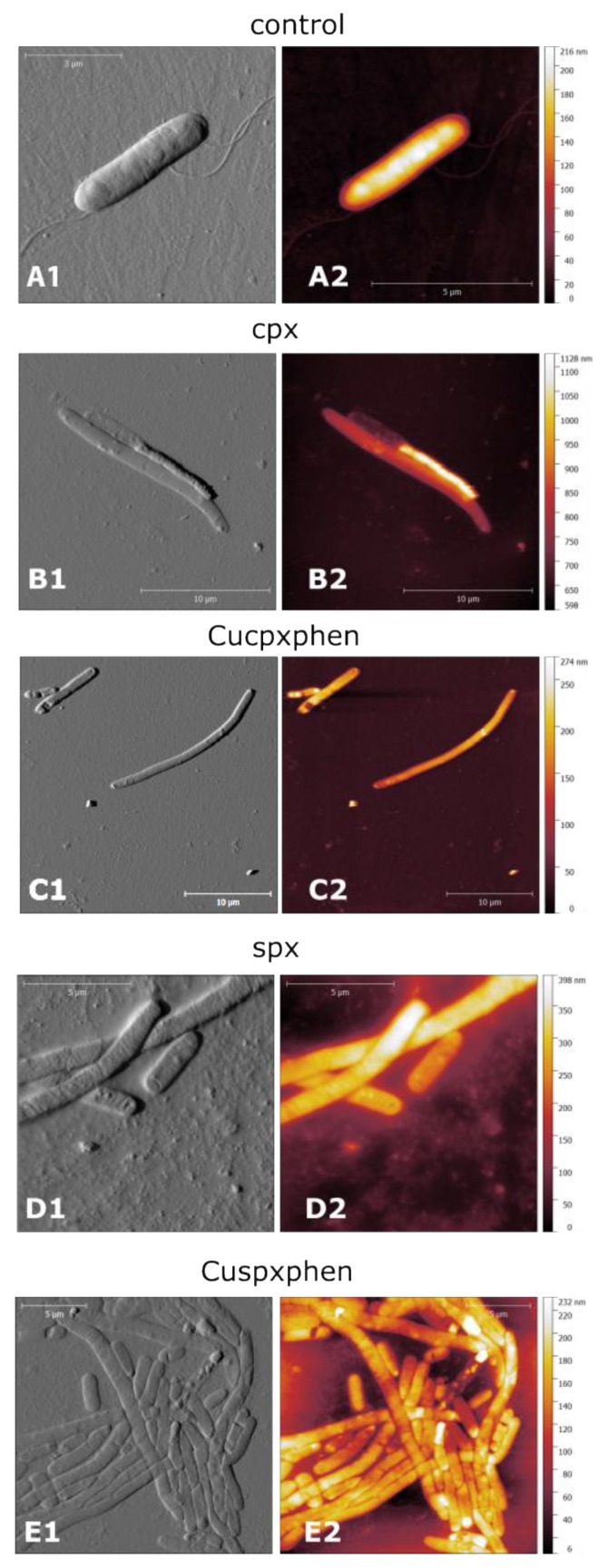
AFM images of *E. coli* ATCC 25922–control cells (**A**) and cells treated with cpx (**B**), Cucpxphen (**C**), spx (**D**) and Cuspxphen (**E**). A1, B1, C1, D1 and E1 are amplitude images; A2, B2, C2, D2 and E2 are height images. These images are representative of the multiple areas from at least three samples analyzed for each condition tested.

**Table 1 ijerph-17-03127-t001:** Minimum inhibitory concentration (MIC) values of several FQs, CuFQphen complexes, phen, Cu(II)/phen (1:1) and Cu(NO_3_)_2_.3H_2_O salt against reference strains, *E. coli* ATCC 25922, *P. aeruginosa* ATCC 27853, *S. aureus* ATCC 25923 and *S. aureus* ATCC 29213. Values presented were obtained from at least three independent experiments.

Compound	MIC Value (μmol dm^−3^)			
	*E. coli* ATCC 25922	*P. aeruginosa* ATCC 27853	*S. aureus* ATCC 25923	*S. aureus* ATCC 29213
cpx	0.012	0.18	0.36–0.75	0.75–1.51
Cucpxphen	0.011	0.17–0.35	0.35–0.71	1.41
erx	0.022	2.78	0.33–0.70	0.33
Cuerxphen	0.022	2.97	0.37–0.74	0.18–0.37
lvx	0.022	1.38	0.33–0.69	0.33
Culvxphen	0.021–0.042	1.39	0.35–0.69	0.35
mxfx	0.018	1.14–2.28	0.07–0.14	0.14
Cumxfxphen	0.019–0.038	2.54	0.08–0.15	0.08–0.15
spx	0.010	0.64–1.27	0.08–0.15	0.15
Cuspxphen	0.005–0.010	0.65	0.32	0.08–0.16
phen	40.4	645.7	80.7–161.4	161.4
Cu(II)/phen (1:1)	72.8	≥1164.1	72.8	145.5
Cu(NO_3_)_2_.3H_2_O	≥4238.2	≥4238.2	≥4238.2	≥3642.2

**Table 2 ijerph-17-03127-t002:** MIC values of several FQs, CuFQphen complexes, phen, Cu(II)/phen (1:1) and Cu(NO_3_)_2_.3H_2_O salt against two MDR isolates of *E. coli* (HSJ Ec002 and HSJ Ec003). Values presented were obtained from at least three independent experiments.

Compound	MIC Value (μmol dm^−3^)	
	HSJ Ec002	HSJ Ec003
cpx	386.3	193.2
Cucpxphen	93.0	45.1–90.3
erx	178.1	178.1
Cuerxphen	95.0	95.0
lvx	22.1	88.6
Culvxphen	44.4	44.4–88.7
mxfx	18.3	18.3
Cumxfxphen	20.3	40.6
spx	40.8	81.6
Cuspxphen	41.6	83.1
phen	80.7	40.4
Cu(II)/phen (1:1)	145.5	72.8
Cu(NO_3_)_2_.3H_2_O	>4238.2	≥4238.2

**Table 3 ijerph-17-03127-t003:** MIC values of several FQs, CuFQphen complexes, phen, Cu(II)/phen (1:1) and Cu(NO_3_)_2_.3H_2_O salt against four MRSA isolates (Sa1-SA3, Sa3-SA3, 19/35 and 26/01). Values presented were obtained from at least three independent experiments.

Compound	MIC Value (μmol dm^−3^)			
	Sa1-SA3	Sa3-SA3	19/35	26/01
cpx	386.3	386.3–772.6	≥3090.5	1545.2
Cucpxphen	90.3	90.3	180.5	180.5
erx	22.3	44.5–89.0	178.1	712.3
Cuerxphen	95.0	95.0	95.0	95.0
lvx	708.4	44.3	177.1	1416.8
Culvxphen	88.7	88.7–177.5	88.7	88.7
mxfx	18.3	292.3	18.3	73.1
Cumxfxphen	10.2	10.2	20.3	40.6
spx	652.4	20.4	163.1	652.4
Cuspxphen	83.1–166.2	41.6	41.6	83.1
phen	2582.9	645.7	322.9	322.9–645.7
Cu(II)/phen (1:1)	72.8	72.8	145.5	145.5
Cu(NO_3_)_2_.3H_2_O	≥4238.2	≥4238.2	≥3642.2	≥3642.2

**Table 4 ijerph-17-03127-t004:** Growth inhibition zones caused by Cucpxphen, Cuspxphen, phen, Cu(II)/phen (1:1), Cu(NO_3_)_2_.3H_2_O salt, cpx and amp against two MRSA isolates (Sa1-SA3 and Sa3-SA3). The diameter of the zones of growth inhibition is presented in mm. The values shown were obtained from at least two independent experiments.

Compound	Compound Alone	Compound + cpx	Compound + amp
	MRSA Sa1-SA3		
Cucpxphen	10.5–11	10	12
Cuspxphen	12	11	12
phen	0	0	12
Cu(II)/phen (1:1)	9–11	9	12
Cu(NO_3_)_2_.3H_2_O	0	0	10
cpx (5 µg/disk)	0	-	-
amp (10 µg/disk)	0	-	-
	**MRSA Sa3-SA3**		
Cucpxphen	9–12	12	12
Cuspxphen	13–14	14	14
phen	0	0	10
Cu(II)/phen (1:1)	0	0	10
Cu(NO_3_)_2_.3H_2_O	8–9	9	11
cpx (5 µg/disk)	0	-	-
amp (10 µg/disk)	10	-	-

**Table 5 ijerph-17-03127-t005:** MIC values of Cucpxphen, Cuspxphen, cpx and amp alone or in combination obtained from the checkerboard method. The MIC values and respective FIC index were determined against three clinical isolates (MDR isolate of *E. coli* HSJ Ec002 and MRSA isolates Sa1-SA3 and Sa3-SA3), from two independent experiments.

	MIC Value (μg mL^−^^1^)				∑ FIC
	Alone		In Combination		
Clinical isolate	Cuspxphen	cpx	Cuspxphen	cpx	
HSJ Ec002	32	128	32	128	2 (I)
Clinical isolate	Cucpxphen	amp	Cucpxphen	amp	
Sa1-SA3	64	64	8	32	0.625 (A)
Clinical isolate	Cucpxphen	cpx	Cucpxphen	cpx	
Sa3-SA3	64	256	16	64	0.5 (S)
Clinical isolate	Cucpxphen	amp	Cucpxphen	amp	
Sa3-SA3	64	256	32	128	1 (A)

Interpretation of the interaction effect: synergy (S), ΣFIC ≤ 0.5; additivity (A), 0.5 < ΣFIC ≤ 1; indifference (I), 1 < ΣFIC ≤ 4; antagonism, ΣFIC > 4.

**Table 6 ijerph-17-03127-t006:** Concentrations of cpx, Cucpxphen and Cuspxphen able to inhibit the activity of DNA gyrase and topoisomerase IV of *E. coli* and *S. aureus*. The values presented were obtained from at least three independent experiments.

Assay	Bacterial Enzyme	Concentration of Compound Able to Inhibit the Enzyme/µmol dm^−3^		
		cpx	Cucpxphen	Cuspxphen
DNA gyrase supercoiling inhibition assay	*E. coli*	5	5	5
	*S. aureus*	50	50	50
Topoisomerase IV relaxation inhibition assay	*E. coli*	10	5–10	5–10
	*S. aureus*	10	5	5

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
