# Peer review of "Fluoroquinolone Metalloantibiotics: A Promising Approach against Methicillin-Resistant *Staphylococcus aureus"

_ijerph, 2020, doi:10.3390/ijerph17093127_

Round 1
Reviewer 1 Report
The manuscript "Fluoroquinolone metalloantibiotics: a promising approach against methicillin-resistant Staphylococcus aureus" concerns the development of the updated anti-microbial preparations on the base of fluoroquinolones. The authors present their study and data thoroughly and efficiently.
The manuscript has some redundancy of the presented data, which partially duplicate each other. Therefore:
1. Table 1 and Figure 3 can be moved to supplementary.
2. The concentrations of antimicrobial agents in Tables 2 - 4 should only be given in mkmol/dm3, this is enough. Please, remove the mkg/ml concentrations.
3. To make easier to compare the original antibiotic and its derivative the order of chemical compounds in Tables 2 - 4 should be changed, for example, cpx, followed by Cucpxphen; erx, followed by Cuerxphen; etc.
Reviewer 2 Report
The article entitled "Fluoroquinolone metalloantibiotics: a promising approach against methicillin-resistant Staphylococcus aureus" showed impressive data. The AFM images are quite impressive and good informations for the reader. But only thing i would like to mention to the authors to improve the english of the manuscript. This will flourish the article and will create more interest to the readers.
Reviewer 3 Report
Ferreira et al in the submitted manuscript demonstrate better efficacy of metalloantibiotics compared to pure FQs, especially against MRSA isolates. The authors further demonstrate synergy in the combination of these complexes.
Both metalloantibiotics and FQs were showed to act on topoisomerase IV and DNA gyrase of bacteria with some preference for topoisomerase IVin Gram positive bacteria. Over all the manuscript is well written and worthy of immediate publication.
My only comment is that the authors should expand MRSA at its first use either in the abstract or in the introduction section
